# The C Terminus of the Ribosomal-Associated Protein LrtA Is an Intrinsically Disordered Oligomer

**DOI:** 10.3390/ijms19123902

**Published:** 2018-12-05

**Authors:** José L. Neira, A. Marcela Giudici, Felipe Hornos, Arantxa Arbe, Bruno Rizzuti

**Affiliations:** 1Instituto de Biología Molecular y Celular, Edificio Torregaitán, Universidad Miguel Hernández, Avda. del Ferrocarril s/n, 03202 Elche (Alicante), Spain; marcela@umh.es (A.M.G.); fhornos@umh.es (F.H.); 2Instituto de Biocomputación y Física de Sistemas Complejos, Joint Units IQFR-CSIC-BIFI, and GBsC-CSIC-BIFI, Universidad de Zaragoza, 50009 Zaragoza, Spain; 3Centro de Física de Materiales (CFM) (CSIC-UPV/EHU)—Materials Physics Center (MPC), 20018 San Sebastián, Spain; mariaaranzazu.arbe@ehu.eus; 4CNR-NANOTEC, Licryl-UOS Cosenza and CEMIF.Cal, Department of Physics, University of Calabria, Ponte P. Bucci, 87036 Rende, Italy

**Keywords:** disordered protein, folding, oligomer, ribosomal protein, protein stability

## Abstract

The 191-residue-long LrtA protein of *Synechocystis* sp. PCC 6803 is involved in post-stress survival and in stabilizing 70S ribosomal particles. It belongs to the hibernating promoting factor (HPF) family, intervening in protein synthesis. The protein consists of two domains: The N-terminal region (N-LrtA, residues 1–101), which is common to all the members of the HPF, and seems to be well-folded; and the C-terminal region (C-LrtA, residues 102–191), which is hypothesized to be disordered. In this work, we studied the conformational preferences of isolated C-LrtA in solution. The protein was disordered, as shown by computational modelling, 1D-^1^H NMR, steady-state far-UV circular dichroism (CD) and chemical and thermal denaturations followed by fluorescence and far-UV CD. Moreover, at physiological conditions, as indicated by several biochemical and hydrodynamic techniques, isolated C-LrtA intervened in a self-association equilibrium, involving several oligomerization reactions. Thus, C-LrtA was an oligomeric disordered protein.

## 1. Introduction

The *lrtA* gene from *Synechococcus* sp. PCC702 is known to express a light-repressed protein [1,2]. Further investigations have shown that LrtA is involved in the stabilization the 70S ribosomal particles [3], as well as in cell survival during stress circumstances. LrtA is related with other proteins that take part in ribosome activity. Under environmental stress conditions, protein synthesis is stopped in a down-regulation process. Reduction of protein production involves: (i) Formation of the inactive 100S disome through dimerization of 70S particles [4], implicating the action of some proteins; or (ii) protein-ribosome interactions which involve the canonical ribosomal proteins [5,6]. The family of ribosomal proteins in *E. coli* includes YfiA (also known as PY or RaiA, ribosome associated inhibitor A); and YhbH (also known as HPF, hibernation promoting factor). YfiA likely inhibits translation indirectly, involving 70S particles [7,8]. Alternatively, HPF stops translation by stabilizing 100S dimers [8,9,10]. Most bacteria have one or more homologues related to HPF or YfiA [10]. These homologues can be classified in long HPF, short HPF and YFiA, on the basis of the length of their sequences and the presence of a specific domain. The conserved domain in all of them has the β-α-β-β-β-α fold [5,11], with a β-sheet formed by four strands and two α-helices packed against it. According to its sequence, LrtA from *Synechocystis* sp. PCC 6803 belongs to the long HPF sub-family. We have previously shown that LrtA is involved in self-association equilibria [12], and has chameleonic structural properties. In particular, molecular dynamics (MD) simulations and experimental analyses suggest that the whole LrtA has a solvent-dependent conformation, where the N terminus adopts the β-α-β-β-β-α fold and the C terminus is disordered and compact [12].

In this work, we have studied the conformational preferences of the isolated C-terminal region of LrtA (residues 102-191), C-LrtA. We aimed to test whether: (i) C-LrtA was disordered and collapsed, as suggested by previous MD simulations of the whole LrtA; and (ii) isolated C-LrtA was oligomeric in solution. Characterizing the degree of disorder in proteins or protein domains, and whether this contributes to attaining a quaternary structure, is important to explain their functions; in fact, most of the intrinsically disordered proteins (IDPs) characterized so far are involved in protein-protein contacts [13], and it is essential to establish how specificity is achieved in those interactions. We show here that C-LrtA was disordered and with a strong self-association tendency, as shown by several biochemical, biophysical and hydrodynamic techniques: Blue-native gels, glutaraldehyde cross-linking, iodide quenching, small-angle X-ray scattering (SAXS), size exclusion chromatography (SEC) and isothermal titration calorimetry (ITC). MD simulations of isolated C-LrtA also predicted a disordered conformation, in reasonable agreement with the experiments. Therefore, we proved that: (i) former MD predictions on C-LrtA, based on the whole parental LrtA, were correct; (ii) the isolated domain has a tendency to self-associate; and (iii) the presence of quaternary interactions in C-LrtA did not induce any stable secondary nor tertiary structure, and therefore, C-LrtA was an oligomeric IDP.

## 2. Results

### 2.1. Isolated C-LrtA Was Intrinsically Disordered in Solution

To map the conformational features of C-LrtA in solution, we used NMR, fluorescence, far-UV (circular dichroism) CD and MD simulations. Fluorescence gives us information about the overall environment around the fluorescent residues (C-LrtA has 4 tyrosine residues). Far-UV CD provides information about the percentages of secondary structure. NMR gives further information about the presence of secondary and tertiary structures. Finally, MD simulations give indications on the conformation of isolated C-LrtA in solution and on the local propensity for secondary structure formation along the backbone.

(a) NMR: In the methyl (Figure 1A) and the amide (Figure 1B) regions of the NMR spectrum of C-LrtA, there was no dispersion, i.e., all the amide signals appeared clustered between 8.2 and 8.6 ppm and most of the alkyl chains appeared between 0.8 and 1.0 ppm. Only a small shoulder appeared up-field shifted at 0.7 ppm, indicative of a local conformation around the methyl group of a valine, leucine or isoleucine probably close to an aromatic residue; we hypothesize that this signal could correspond to the polypeptide patch VIYI (residues 173–176, in the numbering of the whole LrtA).

The NMR spectrum showed significant broadening in all the signals. On the basis of the results of the other techniques (see below in this section, and later in Section 2.2), this could be due to the presence of conformational exchange (equilibria) among protein species with different self-associated order. However, given the mobility of the protein (see MD simulations results, below in this section), we cannot exclude that the broadening observed could be due to conformational exchange in a single protein species.

Therefore, we can conclude from the NMR spectrum that C-LrtA was disordered.

(b) Far-UV CD: The far-UV CD spectrum of C-LrtA showed a minimum of around 200 nm and a wide shoulder at 222 nm (Figure 2A), which could be due to the presence of helix or turn-like structures, although the absorbance of aromatic residues (4 tyrosine and 3 phenylalanine ones) at the latter wavelength cannot be ruled out [14,15]. Although we cannot exclude protein adsorption to the cell at the lowest protein concentration used, the spectrum intensity was protein-concentration dependent (in the range of 10 to 20 μM of protein concentration, Appendix A) and its shape did not change. The fact that the other techniques used (see Section 2.2) indicate the presence of oligomeric species suggests that the variations observed in the far-UV CD spectra are due to the existence of self-associated species.

The shape of the spectrum of C-LrtA was characteristic of IDPs [16]. Its deconvolution, by using the algorithms available at the DICHROWEB site [17,18], yielded percentages of 7–8% for α-helix structure, 15–20% for β-turn, 28–44% for β-sheet and 45–48% for random-coil.

In the presence of increasing GdmCl concentrations, the shoulder at 222 nm of the far-UV CD spectrum decreased (Figure 2B, left axis, black circles) at the two concentrations explored (10 and 20 μM). These results suggest that the shoulder was not due to the presence of any well-fixed structure, but rather to flickering helix- or turn-like motifs, or even local conformations of the aromatic residues [14,15] Attempts to fit these data to the linear extrapolation model failed, as they led to thermodynamic parameters with non-physical meaning (i.e., negative values of m or large values of [GdmCl]_1/2_). We further tested the disordered nature of C-LrtA by performing thermal denaturations. We observed a decrease in the ellipticity as the temperature was increased (Figure 2C, left axis, black circles) and therefore we did not observe a sigmoidal co-operative behaviour, as it should be expected for a well-folded globular domain [16,19].

Therefore, we can conclude from the far-UV CD data that C-LrtA was disordered.

(c) Fluorescence: Fluorescence spectra of C-LrtA showed a maximum at 307 nm, corresponding to its 4 tyrosine residues [20,21]. We carried out GdmCl denaturations by following the <λ> (at two different C-LrtA concentrations) after excitation at 280 nm. At both protein concentrations, we observed a linear decrease in the <λ> as the concentration of chemical denaturant was increased (Figure 2B, right axis, red circles). We could not fit these data to the linear extrapolation model, as fitting led to thermodynamic parameters (m- or [GdmCl]_1/2_-values) with non-physical meaning (i.e., either negative values or values higher than the protein concentration explored). A similar linear tendency was observed in thermal denaturations (Figure 2C, right axis, red circles).

Therefore, we can conclude from the fluorescence data that C-LrtA was disordered.

(d) MD simulations: C-LrtA (the sequence present in the wild-type protein, i.e., without the His-tag) was simulated starting from an extended conformation with a radius of gyration *R*_g_ = 64 Å. The protein structure spontaneously collapsed in 15 ns, and for the subsequent 10 ns maintained a size of *R*_g_ = 24 ± 2 Å, in excellent agreement with the value (*R*_g_ = 25 Å) predicted for a tag-free IDP of 90 residues with the sequence features of C-LrtA [22]. As reported in Figure 3, the secondary structure propensity of the protein in the time interval considered was reasonably consistent. In contrast, the conformations sampled at sufficiently large intervals (e.g., every 0.1 ns) were relatively different in terms of their three-dimensional arrangement, due to the dynamics of C-LrtA. Although not being long enough to obtain a complete statistical ensemble of conformations, the simulation was not prolonged to prevent well-known artifacts, due to over-compaction of the protein [23,24], because a small drift was observed in its size (decrease of *R*_g_ was ~ 0.3 Å/ns on average, Figure 3).

The simulation results concurred to indicate that C-LrtA in solution was a very flexible protein with little secondary structure. The percentages of helical/β-structure were in good agreement with the range of those obtained from the deconvolution of far-UV CD spectra, but the corresponding backbone conformations were in all cases local and did not extend for more than a few residues. Interestingly, among the four tyrosine residues of C-LrtA, only Tyr182 (according to the numbering in intact LrtA) was in a region with β-structure propensity, whereas the other three were in random-coil regions and showed a large conformational freedom in the isolated domain.

In summary, taking into account all the data, as concluded from fluorescence, far- UV CD, NMR and MD simulations, C-LrtA appeared disordered in solution.

### 2.2. Isolated C-LrtA Was Involved in Self-Association Equilibria in Solution

To map the hydrodynamic properties of LrtA we used several biochemical, biophysical and hydrodynamic techniques: Blue-native gels; glutaraldehyde cross-linking; iodide quenching; DOSY-NMR (diffusion ordered spectroscopy NMR); SAXS (small-angle X-ray scattering); SEC (size exclusion chromatography) and ITC (isothermal titration calorimetry). We used such a plethora of different techniques to provide an unambiguous evidence of the presence of oligomerization in disordered C-LrtA. It is important to pinpoint, however, that with NMR we shall obtain information about the low-molecular weight species whose overall rotational tumbling is very fast, and then, we shall be able to obtain information only on the monomer and/or dimer species.

(a) DOSY-NMR: The DOSY-NMR measurements of C-LrtA yielded a translational diffusion coefficient, *D*, of (5.0 ± 0.2) × 10^−7^ cm^2^ s^−1^ (Figure 4A). By taking into account the hydrodynamic radius, *R*_S_, of dioxane (2.12 Å), and its *D* under our conditions ((8.53 ± 0.02) × 10^−6^ cm^2^ s^−1^), the estimated *R*_S_ for C-LrtA was 36 ± 4 Å. We can compare this value with that theoretically determined for a polypeptide with the sequence length of C-LrtA (including the N-terminal His-tag). The *R* value for an unsolvated, ideal, spherical molecule can be estimated from [21]: $R^{}=\sqrt[{3}]{3M\bar{V}/4N_{A}\pi }$, where *N_A_* is Avogadro’s number, *M* is the molecular weight of the C-LrtA construct (12,449.89 Da), and $\bar{V}$ the specific volume of C-LrtA construct (0.721 mL/g). The calculated radius for C-LrtA is 15.3 Å, but taking into account the water shell [21,26], the hydration radius is 18.5 Å; this value is different from that obtained from experimental DOSY measurements. The *R*_s_ for a spherical, folded protein is given by [27]: $R_{S}=(4.75\pm 1.11)N^{0.29}$, where *N* is the number of residues; in a 109-residue-long protein, such as C-LrtA (the His-tag and the 90-residue-long domain), this expression yields 18 ± 4 Å, in good agreement with the other theoretically calculated value. On the other hand, for an unfolded polypeptide chain, the *R*_S_ could be estimated from [27]: $R_{S}=(2.21\pm 1.07)N^{0.57}$; for C-LrtA, the value is 32 ± 15 Å, which is closer to the values measured in the DOSY-NMR experiments; however, the use of that expression yields a value slightly higher (43 ± 15 Å) for a dimeric species. Therefore, by the DOSY-NMR experiments we are only detecting low-molecular weight species, which seemed to be unfolded.

(b) SEC: Different amounts of C-LrtA were loaded in an analytical Superose 12 10/300 GL column at pH 8.0 (50 mM Tris) and 0.250 M NaCl. The chromatograms did not show a sole peak (Appendix A), and the smaller the concentration of the protein the more were the peaks that appeared. This finding, given protein purity (Appendix A), could be attributed to protein-column interactions of some species, which eluted at larger volumes than expected from their size. Similar delayed peaks, due to protein-column interactions, have been observed in the intact LrtA [12]. It is important to note that a small peak appearing at 16.48 mL was also present in the chromatogram of the most concentrated sample (500 μM) (Appendix A), as well as at any other protein concentration; we interpreted this peak as due to a monomeric species interacting with the column.

The elution volumes of one of the peaks showed a hyperbolic dependence as the concentration of protein was changed (Figure 4B). At very high concentrations (500 μM), the protein had elution volumes of 11.98 mL (obtained as the mean of three different measurements, although the elution peak was very broad). This value would correspond to a molecular weight of 100 kDa (for a comparison, a protein, such as ferritin, with a molecular weight of 400 kDa, elutes in the column at 10.11 mL, Appendix A); these results suggest that C-LrtA was probably an octamer under these conditions. On the other hand, at 30 μM, C-LrtA eluted at 13.88 mL, which would correspond to a molecular weight of 31.6 kDa, close to the expected molecular weight of a dimeric species. Therefore, in the column matrix the protein behaved as a self-associated species with several oligomerization orders, depending on the concentration used.

(c) BN-PAGE (blue native polyacrylamide gel electrophoresis): C-LrtA exhibited two species in these experiments, which corresponded to different self-associated species (Appendix A). The protein species in the fastest migrating band corresponded to an apparent molecular weight of 33 kDa (close to the molecular weight of a dimer, and similar to that observed at the most diluted protein concentration in the SEC experiments). On the other hand, the other band corresponded to an apparent molecular weight of 66 kDa, denoting a pentamer. Our results also suggest that increasing the amount of SDS (well-below the concentration used in denaturing SDS-PAGE gels: 33 mM [28]) had significant effects on the population of self-associated C-LrtA species: the larger the proportion of SDS, the higher the amount of self-associated species detected (the critic micellar concentration of SDS is 1.33 mM). It is important to indicate, at this stage, that the BN-PAGE technique can lead to overestimation of the molecular weights, as some proteins can bind Coomassie dye [29].

(d) Glutaraldehyde cross-linking: To detect the presence of oligomeric species in C-LrtA, we also used the glutaraldehyde agent. We observed dimers (close to the band of the protein marker at 32 kDa) (Appendix A) at shorter times after addition of the cross-linking agent, and other high-molecular-weight species at the top of the SDS-PAGE lanes. The population of these high-molecular weight self-associated species increased at the largest incubation times (Appendix A).

(e) KI quenching: It is reasonable to assume that if the self-associated species form at low C-LrtA concentrations, and the tyrosine residues were involved in the association interfaces, then we should be able to follow protein self-association by KI quenching, and we should expect a decrease in the *K*_sv_ constant as the C-LrtA concentration was increased. We observed the following *K*_sv_ values in C-LrtA: 1.5 ± 0.3 M^−1^ (at 5 μM of protein); 1.1 ± 0.2 M^−1^ (at 20 μM of protein); and 1.04 ± 0.05 M^−1^ (at 40 μM of protein, all of them in protomer units). Then, there was a protein-concentration behaviour in the 5–40 μM concentration range for the *K*_sv_ and C-LrtA self-associates.

(f) ITC experiments: We also tried to test whether C-LrtA dissociated upon dilution, using the heat evolved in the reaction monitored by ITC. For experiments performed at a high protein concentration stock, the heat released upon dilution of the protein into the calorimetric cell was consistent with a dissociation reaction for all injections (Appendix A). We tried to fit the heat released to a simple dimer-monomer equilibrium, but the results of the fitting indicated that this assumption was not good enough, suggesting the presence of higher-order equilibria, as indicated by SEC results (see above in this section).

(g) SAXS experiments: The experiments with C-LrtA indicate a *R*_g_ ≈ 26 Å with a υ ≈ 0.33, close to a compact species value, but with a value of *R*_g_ larger than that of a well-folded protein (Section 4, Figure 4C), which is within the range observed for unfolded polypeptide chains [27]. We obtained a good agreement with the expected Guinier regime at low *Q* values, indicating that, although the protein was self-associated, the size of the C-LrtA species was relatively small.

## 3. Discussion

The structural propensities and association features of IDPs and disordered protein domains are still poorly understood compared to those of well-folded protein regions. The difference is especially important for proteins, such as LrtA, which is formed by two distinct domains (i.e., N-LrtA and C-LrtA) with roughly the same sequence length, but with completely distinct conformational features. In a previous work [12], we have hypothesized that N-LrtA has a distinct folding topology that is in common with other members of the HPF protein family [5,11], whereas C-LrtA was predicted to be unfolded. In the present work, we have tested that hypothesis and we have found evidence that isolated C-LrtA is an IDP. The intrinsically disordered nature of C-LrtA was suggested by several pieces of evidence: (i) the lack of dispersion in NMR spectra (Figure 1); (ii) the shape (Figure 2A) and the deconvolution of far-UV CD spectrum; (iii) the absence of all-or-none co-operative transitions in the thermal and chemical denaturations (Figure 2B,C); and (iv) a model of the structure without the His-tag obtained with MD simulation (Figure 3).

It could be thought that the observed C-LrtA self-association is non-specific, that is, the protein has solvent-exposed hydrophobic patches which induce highly unspecific, self-association. However, there are two pieces of evidence that suggest that association was not the result of random solvent-exposed hydrophobic residues. First, the fact that Tyr residues were implicated in the oligomerization indicates that only regions containing those residues were involved. Second, if self-association was to be unspecific very large high-molecular species should be observed in some of the techniques used; in contrast, the highest molecular-weight species is observed to be an octamer (in SEC experiments) and glutaraldehyde cross-linking showed the presence of higher molecular-weight species only at very long incubation times (Appendix A). Finally, it is interesting to note that we have shown that the intact protein, which also self-associates, did not have any large amount of close solvent-exposed hydrophobic patch at physiological pH [12].

Although it is devoid of secondary and tertiary structures, C-LrtA is involved in quaternary contacts, involving different orders of self-associated species, as indicated by several of the techniques used. In general, IDPs intervene in protein-protein contacts [13], but only a few are reported as self-associated species in solution, while keeping their disordered nature. In C-LrtA, some of the four tyrosine residues were involved in its self-association, as judged by the changes in the quenching parameters as the concentration of protein was increased. In the parental, whole protein, tyrosine residues were already found to participate in the oligomerization interface [12]. With the new results in hand, we suggest that some of the four tyrosine residues in C-LrtB were responsible for the self-oligomerization of the whole protein. Then, whereas tyrosine residues in N-LrtA (residues 1–101) seem to contribute to the rigidity of the β-sheet scaffold, tyrosine amino acids in C-LrtA (residues 102–191) seem to be responsible for the quaternary structure of the intact protein. It is important to note that the detected order of the self-associated species in C-LrtA varied among the different biochemical and hydrodynamic techniques used (as it happens for the whole LrtA [12]) indicating the presence of different oligomerization equilibria. That is, the self-association did not involve the simple dimer-monomer equilibrium, but rather a sequence of different order equilibria.

The exact biological significance of self-association in LrtA remains unknown, and it is still a matter of debate. In fact, in spite of similar structural organization and high sequence homology among the members of the HPF family, their functions during stress responses are very different in the organisms to which they belong to [4,7,8,9,10,30]. There are some examples of oligomeric HPFs reported in the literature: for instance, the short *Vibrio choleras* HPF is a dimer, whose dimerization occurs through Zn ions at one of the β-strands of the β-α-β-β-β-α fold; however, it is not known if such dimerization is due to the crystallization process [11]. On the other hand, the HPF of *Staphylococcus aureus* is also a member of the long HPF family, but its C-terminal region is shorter (60-residue long) and it is folded [5]. The protein is a dimer, and its C-terminal region is responsible for this dimerization; furthermore, interactions of the dimeric C-terminal region with ribosomes are responsible for ribosome dimerization. Thus, we hypothesize that in LrtA the oligomeric, disordered C-LrtA domain might be responsible for the dimerization of 70S particles (whose abundance in the cell has been associated with the presence of LrtA [3]) during stress conditions. The fact that C-LrtA is disordered (in contrast to that of *Staphylococcus aureus* HPF [5]) could provide the advantage that LrtA may be bound not only to the 70S particles, but also to other macromolecules, regulating several cyanobacterial processes triggered by stress conditions.

Evidence of the possibility of self-association and supra-molecular order in IDP are starting to be mounting. One of the first reported examples of oligomeric IDPs was the “fuzzy” dimer formed by the cytoplasmic domain of the T-cell receptor zeta subunit [31,32], although this putative dimer was later shown to be a monomer under a wide range of conditions, and its oligomerization was attributed to non-ideal protein-column-resin interactions [33]. However, other studies have used a plethora of biophysical and biochemical techniques to unambiguously show the existence of intact self-associated IDPs. For instance, there are reports describing oligomeric plant IDPs [34]; dimeric proteins in the disordered *umu*D gene products [35]; oligomeric mitochondrial IDPs [36]; oligomeric IDPs which bind to the Polycomb complex [37]; disordered, oligomeric acid-rich proteins of rod photoreceptors [38]; and disordered oligomeric oncogen products [39]. Many of these homoligomeric interactions are deposited in the MFIB database (http://mfib.enzim.ttk.mta.hu) [40], together with others involving hetero-oligomer assemblies, where in all cases mutual folding of the IDP chains occurs. The presence of such a quaternary (homo or hetero-oligomeric) structural organization for an IDP has been explained as due to multiple transient interactions or long-range contacts, which yield a fuzzy self-associated species [41,42,43], involving different degrees of organization in the association process [44].

## 4. Materials and Methods 

### 4.1. Materials

Deuterium oxide, d_11_-Tris acid and isopropyl-β-d-1-tiogalactopyranoside (IPTG) were purchased from Apollo Scientific (Stockport, UK). DNase, kanamycin, the Trizma base and its acid, sodium trimethylsilyl (2,2,3,3-^2^H_4_) propionate (TSP), imidazole, and the His-Select HF nickel resin were from Sigma-Aldrich (Madrid, Spain). The β-mercaptoethanol (β-ME) was provided by BioRad (Madrid, Spain). The protein marker, PAGEmark Tricolor, and Triton X-100 were from VWR (Barcelona, Spain). The protein gel-filtration-column calibration markers were from GE Healthcare (Barcelona, Spain). Dialysis tubing was from Spectrapor (Spectrum Laboratories, Shiga, Japan). Amicon centrifugal devices with a cut-off molecular weight of 3000 Da were from Millipore (Barcelona, Spain). The rest of the materials used were of analytical grade. Water was deionized and purified on a Millipore system.

### 4.2. Protein Expression and Purification

The C-LrtA region (residues 102-191 of LrtA) was cloned by NZytech (Lisbon, Portugal) in a pHTP1 *E. coli* expression vector (between XhoI and NcoI sites), with kanamycin resistance. The final construct contained an N-terminal His-tag to allow for purification (MGSSHHHHHHSSGPQQGLR), and had the overall sequence: MGSSHHHHHHSSGPQQGLRQHGNVKTSEIVEDKPVEENLIGDRA PELPSEVLRMKYFAMPPMAIEDALEQLQLVDHDFYMFRNKDTDEINVIYIRNHGGYGVIQPHQAS.

Expression of C-LrtA was carried out in the *E. coli* BL21 (DE3) strain (Novagen, VWR, Barcelona; Spain) strains with a final kanamycin concentration of 50 mg/mL. We used 1 L flasks to culture the cells. The expression of the protein was induced with a final concentration of 1.0 mM IPTG, when the absorbance observed for the cell culture was 0.4–0.9 at 600 nm, and the growth of the cells continued for 15–16 h at 25 °C 15–16 h at 25 °C (this temperature was chosen to decrease the possibility of aggregates in C-LrtA, as the parental LrtA had a tendency to form inclusion bodies). We harvested the cells in a JA-10 rotor (Beckman Coulter) for 15 min at 8000 rpm. We re-suspended the cellular pellet from 5 L of culture in 50 mL buffer A (500 mM NaCl, 5 mM imidazole, 20 mM Tris buffer (pH 8), 0.1% Triton X-100 and 1 mM β-ME) and adding a tablet of Sigma Protease Cocktail EDTA-free. In the first attempts to get the protocol of purification, we added 2 mg of DNase (per 1 L of culture). We incubated the mixture for 10 min, with gentle agitation in the fridge (4 °C). Next, we sonicated the mixture (by using a Branson sonicator, 750 W), with 10 cycles of 45 s each at 55% of maximal power output, with intervals of 15 s between the cycles, and always keeping the cells in ice. We separated the supernatant by centrifugation at 18,000 rpm for 40 min at 4 °C in a Beckman JSI30 centrifuge with a JA-20 rotor. C-LrtA was present in the supernatant, and we purified it by immobilized affinity chromatography (IMAC), by adding 5 mL of Ni-resin previously equilibrated in buffer A. The mixture was incubated for 20 min at 4 °C, and afterwards, the lysate was separated from the resin by gravity. The washing step was carried out with 20 mL of buffer B (20 mM Tris buffer (pH 8.0), 500 mM NaCl, 1 mM β-ME, and 20 mM imidazole); the protein was eluted by gravity from the column with buffer C (20 mM Tris buffer (pH 8.0), 500 mM NaCl, 1 mM β-ME, and 500 mM imidazole). The solution was dialyzed against 50 mM Tris buffer (pH 8.0). The protein was further purified in a Hi-Trap Mono Q (GE Healthcare) column by using a gradient step from 0 to 1 M NaCl (50 mM Tris buffer, pH 8.0) in 60 min and a flow of 1 mL/min, in an AKTA Basic system (GE Healthcare), while monitoring the absorbance at 280 nm. This column purification step was used based on the theoretical isoelectric point of C-LrtA (pI = 5.72).

In the initial purification attempts (carried out in the presence of DNase), we observed that the main peak coming from the Hi-Trap Mono Q column did not show an emission fluorescence spectrum expected for a protein containing only 4 tyrosine residues (i.e., with a maximum at 308 nm), but rather the spectrum of a protein containing tryptophan (a maximum at ~330 nm). Therefore, we thought of the possibility of contamination of the protein with the DNase used in the first steps of the purification, which also has a similar pI (5.2), and it has a tendency to self-associate. Then, in subsequent purifications we did not add DNase to the cell lysate. As a consequence, the protein coming from the Hi-Trap Mono Q showed absorbance at 260 nm, probably due to the presence of oligonucleotides resulting from the sonication step; these oligonucleotides must be present in a small amount, since no evidence of sharp peaks (contaminants of low-molecular weight) were observed in the 1D ^1^H-NMR spectrum (Figure 1). The presence of traces of oligonucleotides is not infrequent in proteins of the same family as it has been also observed in the recombinant HPF from *S. aureus* after its purification [5]. After elution from the Hi-Trap Mono Q column, the sample was extensively dialyzed against water, and no precipitation was observed in the dialysis tubing. The concentration of the protein Pc (mg/mL) was calculated by [45]: *P*_c_ = 1.55 A_280_–0.75 A_260_, where A_280_ and A_260_ are the absorbances of the dialyzed solution of the protein at 280 and 260 nm, respectively.

### 4.3. Fluorescence

Fluorescence spectra were collected on a Cary Varian spectrofluorimeter (Agilent, Santa Clara, CA, USA), interfaced with a Peltier, at 25 °C. The C-LrtA concentrations used were 5 and 10 μM of protein (in protomer units) in the GdmCl denaturations carried out at pH 7.5 (50 mM Tris buffer). A 1-cm-pathlength quartz cell (Hellma; Sigma Aldrich, Madrid, Spain) was used. In the denaturation experiments with GdmCl, we prepared the samples the day before starting from a 7 M GdmCl concentrated stock that equilibrated overnight; samples were left at 25 °C for 1 h before performing the measurements.

The emission intensity weighted average of the inverse wavelengths (also called the spectrum mass centre, or the spectral average energy of emission), <λ>, was calculated as described [46].

(a) Steady-state spectra: The experimental set-up used in the case of the denaturation experiments with GdmCl was as previously described [46]. In brief, excitation of the protein samples was at 278 nm, and excitation and emission slits were 5 nm in all cases. The experiments were recorded between 300 and 400 nm. The signal was acquired for 1 s and the increment of wavelength was set to 1 nm.

(b) Thermal denaturations: Thermal denaturations of isolated C-LrtA were carried out with the same experimental set-up previously described [46]. These experiments were performed at constant heating rates of 60 °C/h, with an average time of 1 s. Thermal scans were collected at 308 nm after excitation at 278 nm from 25 to 90 °C and acquired every 0.2 °C. Protein concentration was 10 μM (in protomer units).

(c) Fluorescence quenching: Quenching by iodide was examined with concentrations ranging from 5 to 40 μM (in protomer units) at pH 7.0 (phosphate buffer, 50 mM). The experimental set-up for KI was the same described above for the intrinsic fluorescence experiments. The data were fitted to [47]: $F_{0}/F=1+K_{sv}\left[KI\right]$, where *K*_sv_ is the Stern-Volmer constant for collisional quenching; *F*_0_ is the fluorescence intensity in the absence of KI; and *F* is that at any KI concentration. The range of KI concentrations explored was 0–0.7 M. Fittings to the above equation were carried out by using Kaleidagraph (Synergy software).

### 4.4. CD

The far-UV CD spectra were recorded at 25 °C on a Jasco J815 spectropolarimeter (Jasco, Easton, MD, USA City, Japan) equipped with a thermostated cell holder, and interfaced with a Peltier unit. A periodical calibration was performed with (+)-10-camphorsulphonic acid. We used two concentration values for C-LrtA (10 and 20 μM, in protomer units) to check for concentration-dependence of the shape and intensity of spectra, performed at pH 7.0 (50 mM, phosphate buffer). Molar ellipticity was determined as previously indicated [46].

(a) *Steady-state spectra*: Experiments were performed using the experimental set-up described previously [46]. Spectra were corrected by subtracting the baseline in all cases. Protein concentration was 10 and 20 μM (in protomer units) for GdmCl-denaturation experiments.

(b) Thermal denaturations: Experiments were carried out with the same experimental set-up described previously [46] and a protein concentration of 10 μM (in protomer units). Briefly, thermal denaturations were performed at a constant heating rate of 60 °C/h from 25 to 85 °C, a response time of 8 s, a band width of 1 nm, acquired every 0.2 °C, and following the ellipticity at 222 nm.

### 4.5. NMR Spectroscopy

The NMR experiments were acquired at 20 °C on a Bruker Avance DRX-500 spectrometer equipped with a triple resonance probe and z-pulse field gradients. All spectra were processed and analysed by using TopSpin 2.1 (Bruker GmbH, Karlsruhe, Germany). We used TSP as the external chemical shift reference [26].

(a) 1D ^1^H-NMR experiments: The 1D-^1^H-NMR spectrum were acquired with a C-LrtA concentration of 120 μM (in protomer units) in 0.5 mL, 50 mM d_11_-Tris buffer (pH 7.2) in H_2_O/D_2_O (90%/10%, *v*/*v*), without any correction for deuterium isotope effects. The spectrum was acquired with 16 K data points. We acquired 2 K scans with a 6000 Hz spectral width (12 ppm), by using the WATERGATE sequence [48]. Before processing the data, baseline correction and zero-filling were applied.

(b) Translational diffusion measurements: The DOSY experiments of C-LrtA at pH 7.2 were performed with the pulse-field gradient (PFG) spin-echo sequence, as described previously [12,46], with sixteen gradient strengths ranging linearly from 2% to 95% of the total power of the gradient unit. The intensity of the methyl signals, *I*, was fit to: $\frac{I}{I_{0}}=-\exp\left(D\gamma_{H}^{2}\delta^{2}G^{2}\left(\Delta -\frac{\delta }{3}-\frac{\tau }{2}\right)\right)$, where *I*_0_ is the maximum peak intensity of the methyl resonances at the smallest gradient strength; δ is the duration (in s) of the gradient (2.7 ms); *G* is the gradient strength (in T cm^−1^); Δ is the time (in s) between the gradients (150 ms); γ_H_ is the gyromagnetic constant of the proton; and, τ is the recovery delay between the bipolar gradients (100 μs). Samples were exchanged in D_2_O buffer (50 mM d_11_-Tris buffer, pH 7.2, not corrected for isotope effects) by using Amicon centrifugal devices for 4 to 6 h.

### 4.6. Blue-Native PAGE (BN-PAGE)

BN-PAGE was performed in linear 4% to 16% (*w*/*v*) polyacrylamide-gradient gels [28,49,50]. Before running the BN-PAGE, sample aliquots (10 μL) containing 20 μg of C-LrtA were mixed with 1 μL of 5% Coomassie Brilliant blue G stock solution in 750 mM aminocaproic acid. Electrophoresis was initiated at 85 V for 30 min, and then continued at 200 V for 2.5 h, at 4 °C. After electrophoresis, the gels were stained overnight with colloidal Coomassie Blue G 250 [51]. In these gels, it is the negative charge from the Coomassie Blue, capable of binding to the hydrophobic protein surfaces, what determines the electrophoretic mobility. The technique is a method of choice to study the organization of protein complexes in their native state [28,49,50,51].

### 4.7. Glutaraldehyde Cross-Linking

Glutaraldehyde cross-linking was carried out at pH 7.2 (50 mM, Tris buffer). The volume of the sample was 200 μL. The C-LrtA concentration was 120 μM (in protomer units). The protocol used has been described previously [12], and experiments were repeated twice.

### 4.8. SEC

SEC experiments were performed at pH 8.0 (50 mM Tris) and 0.250 M NaCl in a Superose 12 10/300 GL column, which was connected to an AKTA FPLC (GE Healthcare, Barcelona, Spain); absorbance at 280 nm was monitored during the elution. Flow rate was 0.8 mL/min, and C-LrtA was loaded in a volume of 100 μL at several protein concentrations, in the range of 30–500 μM (in protomer concentration). The markers used to calibrate the column were ferritin, catalase, aldolase, albumin, bovine RNase A and blue dextran (GE Healthcare, Barcelona, Spain); the isolated protein markers were loaded in the column in the above described buffer, at the same flow rate, and three independent measurements were performed with each marker.

### 4.9. ITC

The protocol used during the ITC experiments was that previously described [46]. Briefly, the ITC experiments were carried out by using a VP-ITC instrument (Microcal, Northampton, MA, USA). Before the experiments, C-LrtA was dialysed at 4 °C against water at physiological pH. Dilution ITC experiments involved sequential injections of microliter amounts (10 μL) of a concentrated protein solution (498 μM, in protomer units) into the calorimetric cell (1.4 mL), which initially contained water alone.

### 4.10. SAXS

SAXS experiments were performed on a Rigaku 3-pinhole PSAXS-L instrument, at 45 kV and 0.88 mA. The MicroMax-002+ X-ray Generator Systems includes a module with a microfocus sealed tube source, and an X-ray generator unit producing Cu-Kα transition photons, with λ = 1.54 Å wavelength. Vacuum was maintained both in the flight path and sample chamber. A two-dimensional multiwire X-ray detector (Gabriel design, 2D-2000X) was used as a detector of the scattered X-rays. We obtained the azimuthally averaged scattered intensities as a function of the scattering vector *Q* (where *Q* = 4π(λ)^−1^sinθ, and θ represents half the scattering angle). Silver behenate was used as standard for calibration (reciprocal space). The solutions were filling Boron-rich capillaries with an outside diameter of 2 mm and wall thickness of 0.01 mm. The contribution from the corresponding buffer (measured on the same capillary) was subtracted by applying the proper factors obtained from transmission measurements. The sample-detector distance was 2 m, allowing covering a *Q*-range from 0.008 to 0.2 Å^−1^.

From the intensity scattered at low-*Q* values –in the so-called Guinier regime– we can determine the average gyration radius, *R*_g_, of the protein under several conditions, by using the Guinier law: $I(Q)=A\exp(\frac{{-R}_{g}^{2}Q^{2}}{3})$. The pre-exponential factor, *A*, is determined by the molecule concentration, the scattering contrast and the mass of the macromolecules dispersed in the solution. On the other hand, we can estimate the compaction grade through the scaling exponent υ, which relates to *Q* as $I(Q)\approx Q^{-1/\upsilon }$, in the high *Q*-range here explored. The values of υ are 1/3 for a polymeric chain collapsed into a globule; 0.5 for a random-coil polymer (which is the conformation of a linear polymer chain in Θ-conditions); and 0.6 for a swollen chain in a good solvent (self-avoiding-walk conformation). The form factor of a coil, with scaling exponent υ, is described in terms of the so-called generalized Gaussian coil function, given by the expression [52]: $I(Q)\approx \frac{1}{\nu U^{1/2\nu }}\gamma \left(\frac{1}{2\nu },U\right)-\frac{1}{\nu U^{1/\nu }}\gamma \left(\frac{1}{\nu },U\right)$, where $U=\left(2\nu +1\right)\left(2\nu +2\right)Q^{2}R_{g}^{2}/6$ and $\gamma (a,x)=\int_{0}^{x}t^{a-1}\exp(-t)dt$. From the fits of this function to the experimental data, the value of the radius of gyration can also be obtained.

### 4.11. Molecular Modelling

A model of C-LrtA (without the His-tag) was obtained in MD simulations by using a protocol previously adopted for IDPs [23,24]. In brief, simulations were performed with the GROMACS package [53] starting from a protein model built by using VMD [25], and collapsing it in a brute-force run carried out in the isobaric-isothermal ensemble. C-LrtA was initially in an extended conformation, except for backbone turns in correspondence with proline residues. The protein was centered in a rhombic dodecahedron box with a minimum distance of 1 nm from the edge of the simulation box, and surrounded with explicit water molecules. Amino acid residues were adjusted to mimic neutral pH and Na^+^ counterions were added to obtain an overall neutral molecular system. The AMBER ff99SB-ILDN force field [54] was used for the protein, and the TIP3P [55] model for water. Other simulation conditions, including modelling of the electrostatics and van der Waals interactions, and reference values and coupling times for the thermostat and barostat, were as previously described [56,57].

## 5. Conclusions

The C-terminal region of the cyanobacterial protein LrtA, a member of the HPF family, was a disordered domain that self-associated through a mechanism that involved some of its tyrosine residues. These findings clarify a number of important features of intact LrtA, including demonstrating, in an unambiguous way, the presence of two distinct and structurally different domains in this protein. Moreover, in the absence of studies of the N-terminal region (residues 1-101) our results suggest that the unstructured C-terminal one is the region driving the supramolecular self-organization of the whole protein. This study contributes to clarify the structural and stability features of proteins that interact and modulate the ribosome activity, and especially those belonging to the almost unexplored subfamily of long HPF proteins.

## Figures and Tables

**Figure 1 ijms-19-03902-f001:**
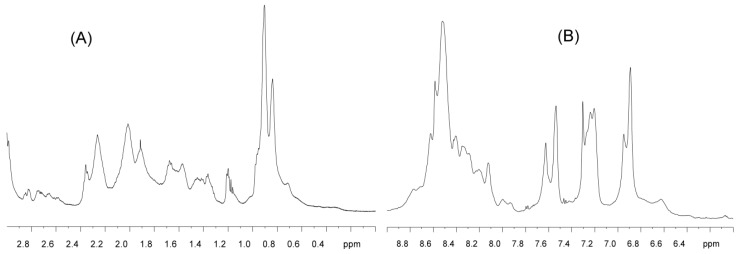
NMR characterization of C-LrtA: (**A**) Methyl and (**B**) amide regions of the 1D ^1^H NMR spectrum of C-LrtA. Spectrum was acquired at 20 °C, and pH 7.2 (50 mM, Tris buffer).

**Figure 2 ijms-19-03902-f002:**
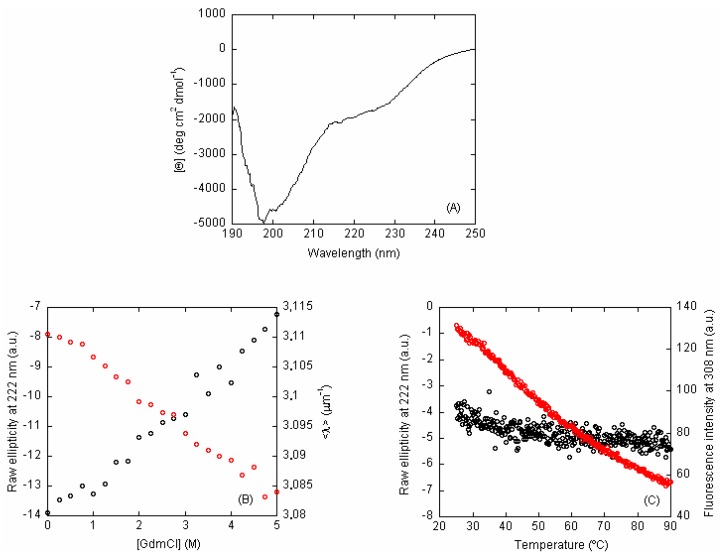
Spectroscopic characterization of C-LrtA: (**A**) Far-UV circular dichroism (CD) spectrum of C-LrtA. Spectrum was acquired at 20 °C, and pH 7.2 (50 mM, Tris buffer) with 20 μM (in protomer units) of protein concentration; (**B**) GdmCl-denaturations of C-LrtA followed by fluorescence (right axis, red circles) and CD (left axis, black circles), at 10 μM of protein concentration (in protomer units) and 20 °C; (**C**) Thermal denaturations of C-LrtA followed by fluorescence (right axis, red circles) and CD (left axis, black circles). Experiments were acquired at pH 7.2 (50 mM, Tris buffer) and 10 μM of protein concentration (in protomer units). The ellipticity (far-UV CD) units of thermal denaturations are arbitrary, because values are scaled up.

**Figure 3 ijms-19-03902-f003:**
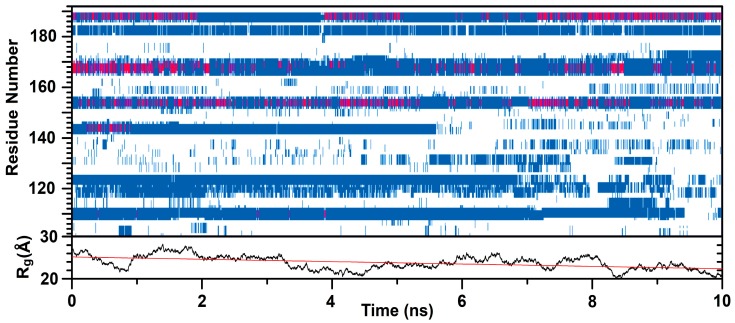
Simulated secondary structure and radius of gyration of C-LrtA: Backbone properties of the protein without the His-tag are calculated in a 10 ns time interval, following 15 ns of equilibration after starting from an elongated conformation. (**Up**) Secondary structure propensities calculated with VMD [25]: (Blue) β-structure, (red) helical structure, and (white) random coil; (**Down**) Radius of gyration of C-LrtA; the drift leading to a small decrease of *R_g_* is also shown (red line).

**Figure 4 ijms-19-03902-f004:**
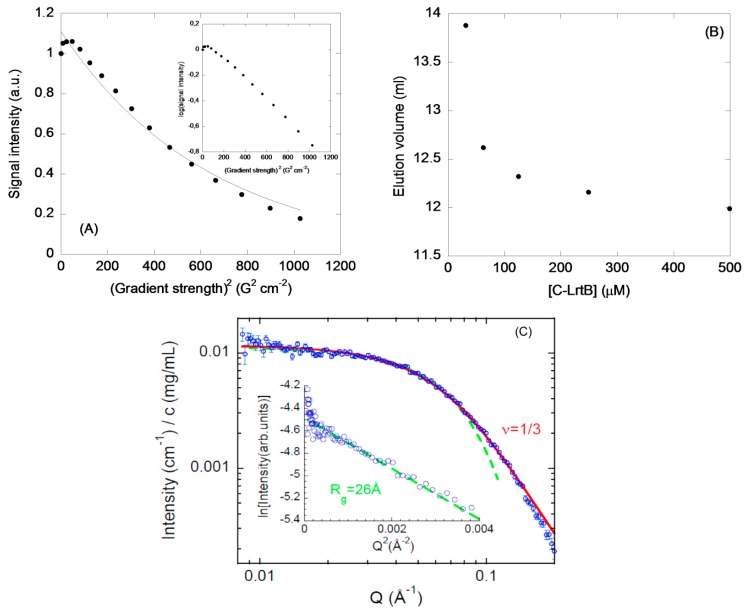
Hydrodynamic and biophysical measurements of C-LrtA: (**A**) DOSY measurements: Intensity decay (arbitrary units) of the methyl signals as the pulse field gradient strength was increased (x-axis). The line is the fitting to equation, as described in Section 4.6. The inset shows the linear relationship between the logarithm of the intensity and the square of the gradient strength; (**B**) size exclusion chromatography (SEC) measurements: Elution volumes of one of the peaks observed for C-LrtA in a Superose 12 10/300 GL at different protein concentrations in buffer pH 8.0 (50 mM Tris) and 0.250 M NaCl; the data have an error of 0.1 mL as obtained from three independent measurements at each particular C-LrtA concentration; (**C**) small-angle X-ray scattering (SAXS) results of C-LrtA are shown with the solid line representing a fit with a generalized Gaussian coil with a scaling exponent value of 1/3 (Section 4.11), and the dashed line is the Guinier description of the low-Q limit (see Guinier plot in the inset, and Section 4.11).

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
