# Peer review of "The C Terminus of the Ribosomal-Associated Protein LrtA Is an Intrinsically Disordered Oligomer"

_ijms, 2018, doi:10.3390/ijms19123902_

Reviewer 1 Report

The authors investigated structural properties of the isolated C-terminal region of LrtA protein by using several biophysical methods.  They found that this protein is intrinsically disordered under the physiological conditions and that it has high tendency to form oligomers.  However, this reviewer could not find any biological significance in these findings, such as the relationship between the biophysical properties and the physiological function of this protein.  In addition, the experimental results does not seem to be self-consistent.  Most of the results might be explained by assuming that this protein is in disordered state with exposed hydrophobic residues, resulting in high tendency to form non-specific aggregates.  For example, the NMR spectra in Fig.1 exhibit a significant line-broadening, suggesting a formation of higher oligomeric states and/or the presence of chemical exchange between several different oligomeric states.  The difference in the intensity of CD spectra shown in Fig.S1 may be explained by the adsorption of hydrophobic proteins to the surface of quarts cell at lower concentration.

Technical comments:

The results of DOSY experiment should be expressed as the plot of log(intensity) vs (gradient strength)^2, showing the linearity between these values.  Similarly, radius of gyration should be estimated by the Guiner plot, log(Q) vs Q^2.

Author Response

Reviewer #1:

The authors investigated structural properties of the isolated C-terminal region of LrtA protein by using several biophysical methods.  They found that this protein is intrinsically disordered under the physiological conditions and that it has high tendency to form oligomers.  However, this reviewer could not find any biological significance in these findings, such as the relationship between the biophysical properties and the physiological function of this protein.  In addition, the experimental results does not seem to be self-consistent.  Most of the results might be explained by assuming that this protein is in disordered state with exposed hydrophobic residues, resulting in high tendency to form non-specific aggregates.  For example, the NMR spectra in Fig.1 exhibit a significant line-broadening, suggesting a formation of higher oligomeric states and/or the presence of chemical exchange between several different oligomeric states.  The difference in the intensity of CD spectra shown in Fig.S1 may be explained by the adsorption of hydrophobic proteins to the surface of quarts cell at lower concentration.

We fully agree with the reviewer. We had not dared to include any comment on the biological significance of the results in our previous version, but now that the reviewers have asked for a relationship about our findings and the biological significance, we have included a paragraph in the Discussion section (section 3, page 9). We have also modified the Introduction section, as the reviewer has suggested in the questionnaire, to provide additional explanations about the importance of studying the molecular basis of protein-protein interactions in IDPs (section 1, page 2).

We further agree with the reviewer that the effect observed in CD (Fig. S1) could be due to adsorption by the quartz material at the lowest protein concentration explored, but the fact that the other experimental techniques indicate the presence of self-association makes us believe that the effect is real. Keeping this in mind, we have, however, also included a remark on such possible adsorption effect in the Results section (section 2.1, pages 2-3).

We have also commented on the signal broadening of the NMR spectrum in Results section (section 2.1, page 2).

Finally, we do not think that the self-association of this protein is driven by unspecific interactions, and we have now discussed this issue on the basis of two pieces of evidence (Discussion section, section 3, pages 8-9) and of additional evidences provided in reference 12 of the manuscript.

The results of DOSY experiment should be expressed as the plot of log(intensity) vs (gradient strength)^2, showing the linearity between these values.  Similarly, radius of gyration should be estimated by the Guiner plot, log(Q) vs Q^2.

We have now included the DOSY line and the Guinier plot as insets in panels A and C of the new Figure 4, respectively.

Reviewer 2 Report

The authors make a fairly humble claim about the disordered nature of the C-terminal domain of LrtA and its oligomerization propensity, which is sufficiently well supported by a wide range of biophysical data.

For the improvement of the introduction, I would probably mention some examples of fuzzy complexes AND add a couple of sentences why disordered structure and oligomerization propensity may be functionally relevant.

Author Response

Reviewer #2:

The authors make a fairly humble claim about the disordered nature of the C-terminal domain of LrtA and its oligomerization propensity, which is sufficiently well supported by a wide range of biophysical data.

We thank this reviewer for his/her words.

For the improvement of the introduction, I would probably mention some examples of fuzzy complexes AND add a couple of sentences why disordered structure and oligomerization propensity may be functionally relevant.

In our previous version, we had provided that information in the Discussion section. Now, we have improved the Introduction, by adding several sentences describing the biological importance of IDPs and their interactions (section 1, page 2). A new reference has been added and the numbering of the rest has been up-dated. The Discussion section where this was mentioned has also been modified (section 3, page 9).

Reviewer 3 Report

The authors apply a plethora of methods to prove that the C-terminal part of the bacterial LrtA protein is disordered and forms oligomeric complexes. The results are novel. The manuscript seems technically sound to me, although I am a bioinformatician so I am probably not the best in judging  the soundness of experimental work. Still, I am familiar with the methodology that is generally applied to show that a protein is disordered an the authors apply many of those methods in combination.

I find the paper is conscise, well-written and easy to follow.

I have only a few questions/suggestions to the authors:

1) Is there any indication that during the formation of oligomeric states of C-LrtA mutual folding (co-folding) occurs? Did the authors try to investigate if there is a gain in secondary structure content due to oligomerization?

2) In the third paragraph of the discussion the authors list some oligomeric IDPs. Here they could refer to the MFIB database (http://mfib.enzim.ttk.mta.hu/) where there are 140 IDPs deposited that can form different types of oligomers, mostly dimers, through mutual folding of the IDP chains.

3) In the introduction section the authors write about the physiological role of the protein, however in the discussion section they do not try to discuss how their findings on the structural/ oligomerization properties of the protein could add to understanding its function. In my opinion they should add an extra paragraph discussing how this modular arrangement of protein, the disordered nature of the C-terminal part and its propensity for oligomerization could serve the modulation of ribosome activity. Also, it would be interesting to know if there is any indication in the literature/in their data that the N-terminal domain is also prone to oligomerize. Or does oligomerization solely depend on the disordered C-terminal tail? Could they draw a model on how the protein functions based on their findings? Is the stochiometry of 100S dimer binding known? Do we know which part of LrtA binds the ribosome? Could the N-terminal conserved domain serve 70S particle binding while the C-terminus ensuring oligomerizaion, thus keeping two 70S particles together and thereby stabilizing the 100S dimer?

Author Response

Reviewer #3:

The authors apply a plethora of methods to prove that the C-terminal part of the bacterial LrtA protein is disordered and forms oligomeric complexes. The results are novel. The manuscript seems technically sound to me, although I am a bioinformatician so I am probably not the best in judging  the soundness of experimental work. Still, I am familiar with the methodology that is generally applied to show that a protein is disordered an the authors apply many of those methods in combination.

I find the paper is conscise, well-written and easy to follow.

I have only a few questions/suggestions to the authors:

We thank this reviewer for his/her words. We also thank him/her for his/her efforts to understand what we have done.

1) Is there any indication that during the formation of oligomeric states of C-LrtA mutual folding (co-folding) occurs? Did the authors try to investigate if there is a gain in secondary structure content due to oligomerization?

We have not been able to obtain C-LrtA as a monomeric species, probably because its self-association constant is very high. What we were trying to convey is that the protein, even being oligomeric, has not structure, as concluded from any of the spectroscopic techniques (fluorescence, CD and NMR) and the MD simulations. We have added several sentences to pinpoint this conclusion (Results section, section 2, pages 2-3).

2) In the third paragraph of the discussion the authors list some oligomeric IDPs. Here they could refer to the MFIB database (http://mfib.enzim.ttk.mta.hu/) where there are 140 IDPs deposited that can form different types of oligomers, mostly dimers, through mutual folding of the IDP chains.

We thank the reviewer for bringing into our attention that database; we did not know it. We have modified the Discussion section (section 3, page 9) to include the reference to the database; the numbering of the rest of the references has been updated.

3) In the introduction section the authors write about the physiological role of the protein, however in the discussion section they do not try to discuss how their findings on the structural/ oligomerization properties of the protein could add to understanding its function. In my opinion they should add an extra paragraph discussing how this modular arrangement of protein, the disordered nature of the C-terminal part and its propensity for oligomerization could serve the modulation of ribosome activity. Also, it would be interesting to know if there is any indication in the literature/in their data that the N-terminal domain is also prone to oligomerize. Or does oligomerization solely depend on the disordered C-terminal tail? Could they draw a model on how the protein functions based on their findings? Is the stochiometry of 100S dimer binding known? Do we know which part of LrtA binds the ribosome? Could the N-terminal conserved domain serve 70S particle binding while the C-terminus ensuring oligomerizaion, thus keeping two 70S particles together and thereby stabilizing the 100S dimer?

We had not dared to include any biological significance of the results in our previous version, but now that the reviewers have asked for a relationship about our findings and the biological significance, we have included a paragraph in the Discussion (section 3, page 9). The model we suggest (which includes some hypotheses on the interaction with the ribosome) is based on some reasonable assumptions, but of course a number of further investigations will be needed to clarify this point in the future.

Round  2

Reviewer 1 Report

The manuscript was improved remarkably.